# Overcoming Pain and Kinesiophobia: Unlocking the Path to Better Knee Osteoarthritis Rehabilitation

**DOI:** 10.3390/biomedicines13061361

**Published:** 2025-06-01

**Authors:** Andrea Pantalone, Teresa Paolucci, Mirko Pesce, Rocco Palumbo, Alessandro Pozzato, Alice Cichelli, Gabriele Santilli, Mariachiara Zuccarini, Antonia Patruno, Marco Tommasi

**Affiliations:** 1Department of Medicine and Aging Sciences, University of Chieti-Pescara, 66100 Chieti, Italy; andrea.pantalone@unich.it (A.P.); mirko.pesce@unich.it (M.P.); antonia.patruno@unich.it (A.P.); 2Department of Oral, Medical and Biotechnological Sciences (DSMOB), Physical Medicine and Rehabilitation, University G. D’Annunzio of Chieti-Pescara, 66100 Chieti, Italy; alice.cichelli@unich.it (A.C.); mariachiara.zuccarini@unich.it (M.Z.); 3CARES, Physical Medicine and Rehabilitation, University G. D’Annunzio of Chieti-Pescara, 66100 Chieti, Italy; 4UdA-TechLab, Research Center, University of Chieti-Pescara, 66100 Chieti, Italy; 5Department of Psychology, University of Chieti-Pescara, 66100 Chieti, Italy; rocco.palumbo@unich.it (R.P.); marco.tommasi@unich.it (M.T.); 6Telea Electronic Engineering srl, Sandrigo, 36066 Vicenza, Italy; alessandro.pozzato@teleamedical.com; 7Physical Medicine and Rehabilitation, Sapienza University of Rome, 00100 Rome, Italy; gabriele.santilli@uniroma1.it

**Keywords:** kinesiophobia, pain, biopsychosocial approach, rehabilitation, osteoarthitis, inflammation

## Abstract

**Objectives:** Knee osteoarthritis (KOA) rehabilitation aims to assess the impact of pain reduction on kinesiophobia and outpatient welfare, emphasizing the interconnectedness of biopsychosocial factors in the rehabilitative process. **Methods**: The study involved a sample of KOA patients undergoing outpatient physical therapy. Forty patients (*n* = 40), aged 40–88, with acute or chronic knee osteoarthritis (Kellegren-Lawrence staging score I–II–III) were collected in Patients undergoing physical therapy using quantum molecular resonance (QMR) technology. The analysis employed a cross-lagged panel model to examine the relationships between perceived pain, kinesiophobia, and quality of life during the rehabilitative plan. **Results**: Rehabilitation significantly reduced pain levels and kinesiophobia while improving the quality of life for outpatients. The analysis demonstrated that pain reduction had a substantial causal influence on kinesiophobia and life conditions, both immediately following treatment and during follow-up. **Conclusions**: The findings underscore the importance of considering biopsychosocial factors in KOA rehabilitative treatment, highlighting the dynamic interplay between pain perception, kinesiophobia, and quality of life throughout the rehabilitation process.

## 1. Introduction

The biopsychosocial (BPS) approach to medicine was introduced in 1977 by Engel [1]. According to this model, healthcare and overall well-being depend not only on the physical or biochemical aspects of the body (the biomedical model) but also on psychological and social factors [2,3,4,5]. The likelihood of developing certain diseases or pathologies is, thus, shaped by a complex interplay of biological, psychological, and social factors—extending beyond mere physical health [3,6].

The BPS model emphasizes the dynamic relationships among biological, psychological, and social factors, which collectively influence health conditions and psychological well-being [6]. To effectively analyze these interactions, it is essential to determine the contribution of each factor and how they influence one another. These relationships are often conceptualized as dynamic processes, which are typically studied through statistical analyses, such as analysis of variance or correlation studies, rather than deterministic mathematical models [6,7]. Because of the complex, interactive nature of these variables, the BPS approach tends to focus on individual specificity, recognizing that each pathology has a unique history linked to a person’s development and life circumstances rather than categorizing patients into fixed groups [3,6].

This individualized perspective is reflected in the use of multivariate statistical models that account for personal differences, helping to clarify the contribution of various factors to disease etiology and progression. Such models are particularly beneficial in managing chronic conditions or long-term pathologies, like musculoskeletal diseases such as osteoarthritis, where extended rehabilitation is often necessary [8,9].

In particular, knee osteoarthritis (KOA) is a global health concern, especially prevalent among the elderly over 65 years of age [10]. This condition is characterized by moderate to severe pain and disability, which significantly impair quality of life and contribute to rising public health costs [11]. Beyond physical symptoms, psychological factors—such as anxiety, stress, and fear of pain (kinesiophobia)—also play a vital role in shaping patients’ experiences during rehabilitation [12,13].

Kinesiophobia, a component of the fear-avoidance model, is defined as an excessive, irrational fear of movement due to perceived vulnerability to injury or re-injury [12]. This fear can negatively influence the lifestyle of KOA outpatients by fostering beliefs that movement may cause further damage, leading to avoidance behaviors. Such avoidance reduces physical activity, which can result in muscle weakness, joint instability, and ultimately, a vicious cycle of worsening symptoms and decreased autonomy [14,15,16].

Research suggests that rehabilitation can positively impact KOA patients by reducing kinesiophobia, even if their pain levels remain relatively unchanged [17]. However, it remains unclear whether improvements are primarily due to psychological effects, such as decreased fear of movement, or physical benefits, like pain reduction and increased joint stability. Some studies indicate that psychological factors, including fear and anxiety, can diminish with rehabilitation, especially in overweight patients, thereby enhancing overall well-being [17,18,19,20].

Despite these insights, the existing literature on kinesiophobia and osteoarthritis often lacks a cross-lagged or longitudinal perspective. Rehabilitation in KOA is a dynamic process where biological, psychological, and social factors are interconnected, aligning with the BPS model [21].

Our hypothesis considers that kinesiophobia could act as a significant barrier to effective rehabilitation, leading patients to avoid activities that could alleviate symptoms or improve joint function; therefore, it is crucial to examine how these factors influence each other over time. To better understand these complex interactions, our study focused on analyzing how perceived pain (a biological factor), kinesiophobia (a psychological factor), and quality of life (a social and life condition factor) change during a specific rehabilitative treatment for KOA outpatients.

This fear of movement often creates a vicious cycle: avoidance behaviors lead to muscle weakness and joint deterioration, which in turn exacerbate pain and disability. Consequently, addressing kinesiophobia is essential for successful rehabilitation. Effective strategies typically involve educational approaches and therapeutic interventions, such as cognitive–behavioral therapy, personalized exercise programs, and integrated physical therapy. These methods aim to help patients understand that controlled, gradual movement is safe and beneficial, thereby reducing fear and promoting activity.

The primary goal of our study was to assess whether, during rehabilitation, reductions in perceived pain—achieved through physical therapy—also lead to decreases in kinesiophobia and improvements in quality of life. In other words, we aimed to explore the dynamic processes activated by rehabilitation: whether improvements in physical symptoms directly influence psychological factors and social well-being, or if reductions in fear and avoidance behaviors are primarily responsible for enhancing patients’ overall quality of life.

Understanding these interactions is vital for developing more effective, individualized treatment plans. By applying a BPS perspective, our research emphasizes the importance of considering biological, psychological, and social dimensions simultaneously. This comprehensive approach could help identify which factors are most influential at different stages of rehabilitation and how they interact over time, ultimately leading to better management of KOA and similar chronic conditions.

## 2. Materials and Methods

We conducted a longitudinal design model, which is suitable for research in rehabilitation per the TRENDS guidelines [22].

### 2.1. Participants

Forty patients (*n* = 40) from the Clinical Hospital “SS Annunziata” of Chieti (Italy), referred for rehabilitation, underwent physiotherapy with two experienced physiotherapists. Eligibility was determined during clinical consultation by an orthopedic expert in KOA and based on specified criteria: aged 40–88, with acute or chronic knee osteoarthritis and pain (visual analog scale > 3). The inclusion criteria were: both male and female subjects and radiographic evidence for Kellegren–Lawrence staging score (I–II–III). Patients with the following symptoms were excluded: favism, hemolytic anemia, severe hyperthyroidism, graves’ disease, thrombocytopenia < 50,000 and severe coagulopathy, severe cardiovascular instability, coagulation disorders, alcohol abuse, hemochromatosis, patients treated with dietary supplements, pregnancy and lactation, psychiatric disorders, less than three months after previous knee infiltration, septic arthritis and/or febrile conditions, and history of contraindications to current instrumental physiotherapy (previous cancer). Subjects with rheumatic and autoimmune diseases and a recent history of trauma and/or distortions of the knee (ligaments) were excluded from the study. Recruitment spanned six months, with informed consent obtained. No new medications or therapies were introduced during the study. This stringent selection process ensured patients met criteria and maintained stable treatment regimens, enhancing research reliability.

### 2.2. Treatment Rehabilitative Protocol

Physical therapy was performed with the Q-Physio model electro-medical device (code 4001006), serial number D06164121, by Telea Electronic Engineering Srl (Vicenza—Italy). Patients underwent the following treatment by this electrotherapy: three sessions per week, for a total of 6 sessions, each lasting 30 min.

Rehabilitative treatments were performed with the patient in the supine position in the semi-flexion knee, and four electrodes were used: on the supra-patellar area surface, between the medial femoral condyle and the medial tibial condyle, on the area between the lateral femoral condyle and the lateral tibial condyle, and on the popliteal angle surface. The floating electrode was placed between the couch and the patient’s gluteal region to maximize the contact area. The device generates alternating electric currents characterized by high-frequency (4–64 MHz) and low-intensity waves [23,24].

### 2.3. Outcome Measures

Pain and function were assessed using the visual analogue scale (VAS) [25,26], the knee injury and osteoarthritis outcome scale (KOOS) [27], and the Tampa scale of kinesiophobia (TSK) [28,29].

VAS measures pain intensity on a continuous horizontal 10 cm scale with two start and end points marked “no pain” and “worst pain ever”. The higher the distance forms the start point of the line (on the left), and the higher the perceived pain. Values can range from 0 (= no pain) to 10 (= maximum pain).

KOOS, a self-completed questionnaire, gauges knee-related issues across five subscales (frequency and intensity of pain during functional activities; symptoms like stiffness, swelling, presence of joint noise or locking ROM limitation; difficulty with activities of daily living; difficulty with recreational/sports activities; knee-related quality of life). The item score was measured on a Likert scale from 0 (= no difficulty) to 4 (= very difficult). The global score is then transformed to a percentage that can vary from 0 (= highest presence of knee problems) to 100 (= absence of knee problems).

TSK evaluates pain-related beliefs and fear of movement; TSK-13, in the most common version, rates items on a 4-point Likert scale ranging from 1 “strongly disagree” to 4 “strongly agree”. The global score varies from 13 to 52. Higher scores indicate greater fear. TSK-13 categorizes items into activity avoidance (e.g., “I’m afraid that I might injure myself if I exercise”) and somatic focus factors, providing insight into subjects’ perspectives on pain and movement (e.g., “Pain always means I have injured my body”).

### 2.4. Timelines

All patients were evaluated at T0 = before treatment; T1 = at the end of treatment; T2 = one month after the end of treatment (follow-up).

### 2.5. Statistical Analysis

We calculated the descriptive statistics (mean and standard deviation, SD) of measures taken in the three phases T0 (before treatment), T1 (after treatment), and T2 (follow-up).

Cross-lagged panel models (CLPMs) are statistical models analyzing longitudinal data with observations at multiple time points [30]. They estimate relationships between variables over time, distinguishing directional influences and stability. CLPMs use regression to estimate cross-lagged paths (β_1_ and β_2_), between variables, autoregressive paths (β_3_ and β_4_) indicating variable stability, and synchronous correlations (*r*_X0Y0_ and *r*_X1Y1_). Autoregressive coefficients closer to zero suggest high variance. In a basic CLPM with variables X0, X1, Y0, and Y1 variables, equations express the model’s mathematical structure. CLPMs enable an understanding of how variables influence each other over time, which is essential in longitudinal data analysis. The CLPM is mathematically expressed as a set of equations:Y1 = β_1_X0 + β_3_Y0(1)X1 = β_4_X0 + β_2_Y0(2)
and the relative path-diagram is shown in Figure 1.

If β_3_ and β_4_ are significantly different, X and Y are stable. Significant β_1_ and β_2_, with differing values, suggest causal predominance. When β_1_ > β_2_, X0 influences Y1 more than Y0 influences X1; if β_1_ < β_2_, the reverse holds. Testing causal predominance via structural equation models in r compares CLPMs with free (β_1_ ≠ β_2_) versus equal (β_1_ = β_2_) cross-lagged coefficients. If no statistical difference occurs, no causal predominance exists. We tested CLPM using two-wave data (T0, T1, T2), comparing Model M1 with Models M2, M3, and M4. M1 is the model in which every parameter for cross-relations is set free. M2 is the model in which the crossed relation coefficients of the first wave of measurements from T0 to T1 are set equal (β_1_ ≠ β_2_). M3 is the model in which the crossed relation coefficients of the second wave of measurements from T1 to T2 are set equal each (β_3_ ≠ β_4_). M4 is the model in which all the crossed relation is set equal (β_1_ ≠ β_2_ and β_3_ ≠ β_4_).

Each model’s Akaike information criterion (AIC), Bayesian information criterion (BIC), χ^2^, df, p(χ^2^), and χ^2^ differences were reported, determining significance between Model M1 and others (M2, M3, and M4). Models with the lowest AIC and BIC show the best fit. All the analyses were made with JASP 0.18.1 software [31]. CPLM offers advantages over traditional methods of analysis because (a) it allows us to determine whether prior variables predict later variables and vice versa (directionality of relationship); (b) it includes autoregressive paths that control for prior variables (associations between variables are not merely due to individual differences); (c) it allows the analysis of reciprocal influences over time (dynamic interplay between variables).

## 3. Results

For the descriptive statistics, 64.7% of patients were female and 62.7% were unemployed or retired; 80.4% were married and 19.6% were single. The mean age was 64.6 years (SD = 11.1), and the mean BMI was 26.9 (SD = 6.2) (Figure 2, flow diagram).

Table 1 shows the descriptive analysis for VAS, TSK, and KOOS taken in the three moments T0, T1, and T2. Table 1 demonstrates a consistent reduction in pain from T0 to T1 (first wave) and from T1 to T2 (second wave). Moreover, there is a continual improvement in knee functionality and quality of life, evidenced by the increasing KOOS scores, alongside a consistent decrease in kinesiophobia, particularly noticeable in the first wave. Consequently, rehabilitation proves effective in alleviating pain and kinesiophobia while enhancing outpatient quality of life (Appendix A. Shows Regression coefficients, covariances and variances of model M1 between VAS and TSK scores or between VAS and KOOS scores.)

Figure 3 shows the path diagram of the CLPM that relates VAS measures to TSK measures. Significant coefficients and correlations are indicated with asterisks.

Table 2 reports the coefficient estimations for the four models M1, M2, M3, and M4 and ANOVA results of the comparison between the models.

The differences between M2 and M1 and between M4 and M1 are not significant. The difference between M3 and M1 is significant. The significant difference between M3 and M1 indicates that the crossed-relation coefficients of the second wave are not equivalent and that there is a causal predominance of VAS to TSK. This causal predominance is not present in the first wave.

Figure 4 shows the path diagram of the CLPM that relates KOOS measures to TSK measures.

Table 3 reports the coefficient estimations for the four models M1, M2, M3, and M4 and ANOVA results of the comparison between the models. The disparities between M2 and M1, as well as between M4 and M1, are significant. Although the difference between M3 and M1 is not statistically significant, it approaches significance closely. The cross-lagged coefficients of both the first and second waves are not statistically equivalent, indicating a causal predominance of VAS over KOOS. This causal predominance in both waves is notably underscored by the significant difference between M4 and M1.

## 4. Discussion

Pain is a physical sensation that can profoundly impact human well-being, constituting a multidimensional phenomenon as it affects both the social and psychological aspects of outpatients’ lives [32]. Pain undeniably has a substantial effect on the well-being of individuals suffering from articulation diseases like arthritis or osteoarthritis [13,33]. According to certain studies [12,13], the fear of pain can be regarded as an alternative or concomitant cause of the reduction in outpatients’ well-being. The primary explanation lies in the activation of avoidance behavior to prevent the recurrence of pain sensation [16]. Our analysis of causal predominance among pain sensation, kinesiophobia, and quality of life indicates that pain is the predominant cause both in generating kinesiophobia and in reducing outpatient quality of life.

Our research, even though it is subject to limitations such as the absence of a comparator or control group and the short follow-up window, reveals a positive relationship between pain and kinesiophobia, signifying that higher levels of pain are associated with elevated levels of fear of pain. This relationship, present before treatment, persists and strengthens even after treatment. Therefore, it is reasonable to suppose that after treatment, participants are more sensitive to the persistence of pain, leading them to overreact to pain with higher levels of fear, stress, and anxiety.

The relationships between pain and quality of life are positive, indicating that higher levels of pain are correlated with lower quality of life and increased difficulty during daily activities. It is reasonable to assume that before and after treatment, the persistence of pain leads participants to perceive a high level of difficulty in physical movements. Importantly, after treatment, there was a significant reduction in the crossed-relation coefficient between VAS and KOOS (from −2.179 to −1.264). This reduction is likely due to the positive effect of the treatment on the physical condition and pain perception of participants. These research findings demonstrate that pain management interventions should be prioritized early in rehabilitation to prevent or reduce kinesiophobia. Besides that, our results affirm that the rehabilitative process for knee osteoarthritis exhibits a dynamic multivariate pattern of variables in which biopsychosocial factors play a pivotal role [8,21]. Specifically, rehabilitation demonstrates significant efficacy in alleviating pain levels and kinesiophobia while enhancing overall quality of life.

The CLPM reveals that pain reduction exerts a predominant causal influence on kinesiophobia and life conditions, not only immediately following treatment but also during follow-up. Thus, the efficacy of rehabilitation hinges primarily on its ability to alleviate chronic pain, subsequently leading to decreased psychological anxiety and an enhanced quality of life.

Although the anti-inflammatory effects of quantum molecular resonance (QMR) technology, as already demonstrated in an in vitro model of osteoarthritis-related inflammation [20], may be one of the crucial factors in the management of KOA, on the other hand, the treatment of pain and inflammation alone in patients with KOA is not sufficient to reduce fear of movement. In fact, Selçuk et al. suggest that approaches to increase awareness of fear of movement and physical activity and cognitive behavioral therapy related to fear of movement should be included in the treatment program [34].

Based on our suggestion, it would be beneficial to emphasize that early pain-management interventions should be prioritized in rehabilitation to help prevent or reduce kinesiophobia, ultimately supporting better functional outcomes. Regarding potential confounders, such as the wide age range of participants, acknowledging how these factors might influence the rate or effectiveness of fear reduction is important. Future studies could stratify results by age or other variables to better understand these dynamics. Finally, being transparent about the study’s limitations, like the short follow-up period and the lack of a control group, would indeed strengthen the credibility of the findings. Recognizing these constraints invites further research under more controlled conditions, helping to build a more robust evidence base for managing fear of movement in KOA.

## 5. Conclusions

In conclusion, outpatients with pain in KOA experience elevated levels of kinesiophobia and diminished quality of life primarily as a result of their pain sensations. A rehabilitative program by physiotherapy with an electro-medical device enhances their psychological and living conditions by alleviating pain. The persistence of pain post-treatment can significantly influence kinesiophobia, as it may signify ongoing leg articulation issues. Effectively reducing pain not only improves physical function but also decreases fear of movement, which can lead to better psychological and social outcomes. The persistent relationship between pain and kinesiophobia, even during follow-up, suggests that alleviating chronic pain is crucial for reducing anxiety, stress, and ultimately enhancing overall quality of life.

Furthermore, our results emphasize the need for an integrated, biopsychosocial approach, combining pain management techniques with strategies like awareness training and cognitive-behavioral therapy. Such approaches can help patients overcome fear of movement, encouraging active participation in therapy and improving functional outcomes.

We hope that our work enables physiotherapists and all medical and psychological professionals and practitioners in the field of musculoskeletal rehabilitation to enhance the efficacy of their treatments through a deeper understanding of the dynamic relationships among various biopsychosocial factors, such as pain, kinesiophobia, and quality of life, that influence healthcare quality.

## Figures and Tables

**Figure 1 biomedicines-13-01361-f001:**
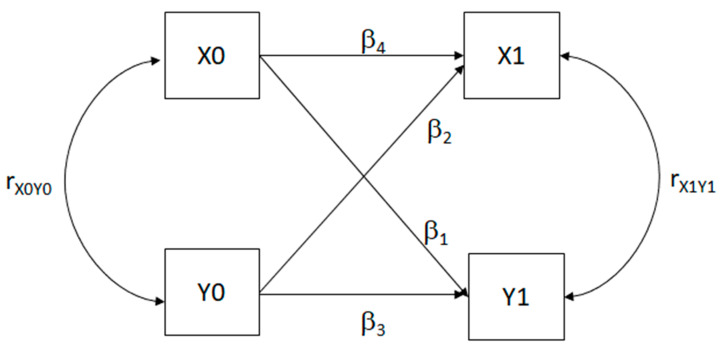
Path diagram of a simple cross-lagged panel model (CLPM). X0 and Y0 = starting measures; X1 and Y1 = consequent measures. β_1_ and β_2_ = coefficient of crossed connections for causal predominance; β_3_ and β_4_ = autoregressive coefficients; r_X0Y0_ and r_X1Y1_ = synchronous correlations.

**Figure 2 biomedicines-13-01361-f002:**
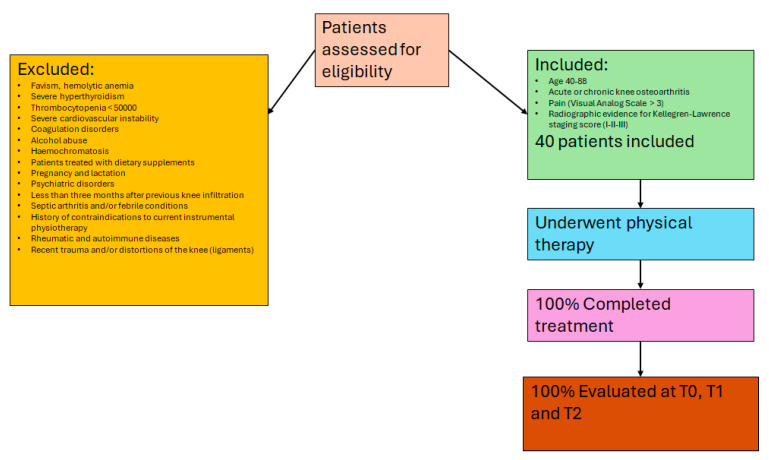
Flow diagram showing patient recruitment, with inclusion and exclusion criteria, and completion rates for each phase of the study.

**Figure 3 biomedicines-13-01361-f003:**
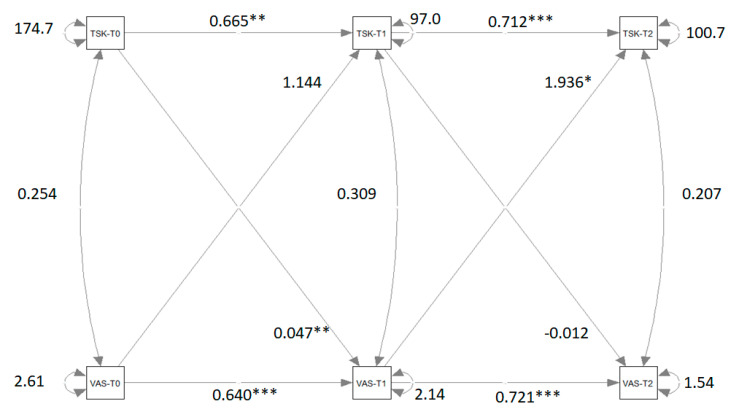
Path diagram of the cross-lagged panel model that relates VAS measures to TSK measures. Significant β coefficients and correlations r are reported with asterisks. Note: VAS = visual analog Scale; TSK = Tampa Scale of Kinesiophobia; T0 = pre-treatment condition; T1 = post-treatment condition; T2 = follow-up. ** = *p* < 0.01; *** = *p* < 0.001. (**** Very significant; ** Moderate significant; * significant*).

**Figure 4 biomedicines-13-01361-f004:**
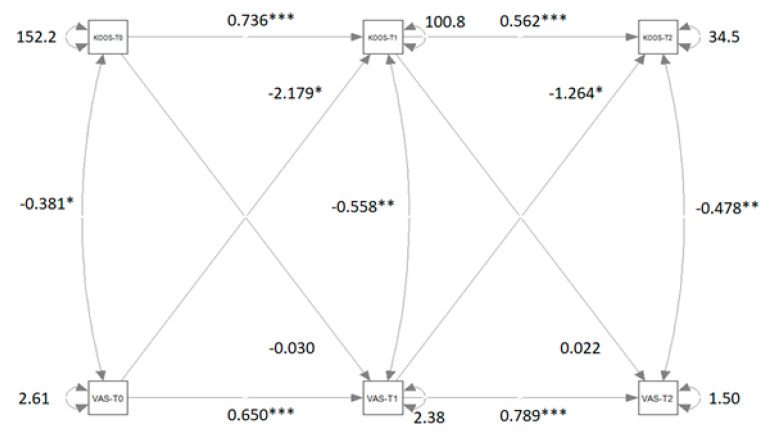
Path diagram of the Cross-lagged panel model that relates VAS measures to KOOS measures. Significant β coefficients and correlations r are reported with asterisks. Note: VAS = visual analog scale; KOOS = knee injury and osteoarthritis outcome score; T0 = pre-treatment condition; T1 = post-treatment condition; T2 = follow-up. ** = significant at *p* < 0.01; *** = significant at *p* < 0.001. (**** Very significant; ** Moderate significant; * significant*).

**Table 1 biomedicines-13-01361-t001:** Descriptive statistics (mean and SD) for VAS, KOOS, and TSK, taken in different time phases (T0, T1, and T2).

Measure	Time Phases	Mean	SD
VAS	T0	5.272	1.636
	T1	3.005	2.007
	T2	2.260	1.853
KOOS	T0	64.850	12.495
	T1	75.800	15.020
	T2	83.425	11.914
TSK	T0	64.700	13.384
	T1	53.900	13.810
	T2	51.550	16.000

Note: SD = standard deviation; VAS = visual analog scale; KOOS = knee injury and osteoarthritis outcome score; TSK = Tampa scale of kinesiophobia. Significant probabilities are in bold type. T0 = pre-treatment condition; T1 = post-treatment condition; T2 = follow-up.

**Table 2 biomedicines-13-01361-t002:** Coefficient estimations for the four models M1, M2, M3, and M4 and ANOVA results of the comparison between the models for VAS and TSK measures. In M1, the coefficients of the two waves (from T0 to T1 and from T1 to T2) are set differently (β_1_ ≠ β_2_ and β_3_ ≠ β_4_); in M2 β_1_ = β_2_ and β_3_ ≠ β_4_; in M3 β_1_ ≠ β_2_ and β_3_ = β_4_; in M4 β_1_ = β_2_ and β_3_ = β_4_.

Model	Predictor	Outcome	Coefficient Estimate	SE	*z*-Value	p(*z*)	AIC	BIC	χ^2^	Model Comparison	χ^2^ Difference	df	p(χ^2^)
M1	VAS-T0	TSK-T1	1.144	0.997	1.148	0.251	1378.5	1417.3	4.653				
	TSK-T0	VAS-T1	0.047	0.018	2.624	**0.009**							
	VAS-T1	TSK-T2	1.936	0.948	2.041	**0.041**							
	TSK-T1	VAS-T2	−0.012	0.017	−0.728	0.466							
M2	VAS-T0	TSK-T1	0.049	0.018	2.739	**0.006**	1377.7	1414.8	5.842	M2 vs. M1	1.189	1	0.276
	TSK-T0	VAS-T1	0.049	0.018	2.739	**0.006**							
	VAS-T1	TSK-T2	1.936	0.934	2.073	**0.038**							
	TSK-T1	VAS-T2	−0.012	0.016	−0.753	0.452							
M3	VAS-T0	TSK-T1	1.144	0.997	1.148	0.251	1380.5	1417.6	8.649	M3 vs. M1	3.996	1	**0.046**
	TSK-T0	VAS-T1	0.047	0.018	2.624	**0.009**							
	VAS-T1	TSK-T2	−0.008	0.017	−0.470	0.638							
	TSK-T1	VAS-T2	−0.008	0.017	−0.470	0.638							
M4	VAS-T0	TSK-T1	0.049	0.018	2.739	**0.006**	1379.7	1415.1	9.838	M4 vs. M1	5.184	2	0.075
	TSK-T0	VAS-T1	0.049	0.018	2.739	**0.006**							
	VAS-T1	TSK-T2	−0.008	0.016	−0.485	0.627							
	TSK-T1	VAS-T2	−0.008	0.016	−0.485	0.627							

Note: SE = standard error; AIC = Akaike information criterion; BIC = Bayesian information criterion; df = degree of freedom. VAS = visual analog scale; TSK = Tampa scale of kinesiophobia. Significant probabilities are in bold type.

**Table 3 biomedicines-13-01361-t003:** Coefficient estimations for the four models M1, M2, M3, and M4 and ANOVA results of the comparison between the models for VAS and KOOS measures. In M1 the coefficients of the two waves (from T0 to T1 and from T1 to T2) are set differently (β_1_ ≠ β_2_ and β_3_ ≠ β_4_); in M2 β_1_ = β_2_ and β_3_ ≠ β_4_; in M3 β_1_ ≠ β_2_ and β_3_ = β_4_; in M4 β_1_ = β_2_ and β_3_ = β_4_.

Model	Predictor	Outcome	Coefficient Estimate	SE	*z*-Value	p(*z*)	AIC	BIC	χ^2^	Model Comparison	χ^2^ Difference	df	p(χ^2^)
M1	VAS-T0	KOOS-T1	−2.179	1.063	−2.050	**0.040**	1311.8	1350.7	6.436				
	KOOS-T0	VAS-T1	−0.030	0.021	−1.419	0.156							
	VAS-T1	KOOS-T2	−1.264	0.636	−1.987	**0.047**							
	KOOS-T1	VAS-T2	0.022	0.018	1.266	0.205							
M2	VAS-T0	KOOS-T1	−0.040	0.021	−1.909	0.056	1313.7	1350.8	10.295	M2 vs. M1	3.859	1	**0.049**
	KOOS-T0	VAS-T1	−0.040	0.021	−1.909	0.056							
	VAS-T1	KOOS-T2	−1.264	0.637	−1.984	**0.047**							
	KOOS-T1	VAS-T2	0.022	0.017	1.330	0.184							
M3	VAS-T0	KOOS-T1	−2.179	1.063	−2.050	**0.040**	1313.7	1350.8	10.262	M3 vs. M1	3.826	1	0.050
	KOOS-T0	VAS-T1	−0.030	0.021	−1.419	0.156							
	VAS-T1	KOOS-T2	0.010	0.017	0.605	0.545							
	KOOS-T1	VAS-T2	0.010	0.017	0.605	0.545							
M4	VAS-T0	KOOS-T1	−0.040	0.021	−1.909	0.056	1315.5	1351	14.121	M4 vs. M1	7.685	2	**0.021**
	KOOS-T0	VAS-T1	−0.040	0.021	−1.909	0.056							
	VAS-T1	KOOS-T2	0.010	0.016	0.630	0.529							
	KOOS-T1	VAS-T2	0.010	0.016	0.630	0.529							

Note: SE = standard error; AIC = Akaike information criterion; BIC = Bayesian information criterion; df = degree of freedom. VAS = visual analogue scale; KOOS = knee injury and osteoarthritis outcome Score. Significant probabilities are in bold type.

## Data Availability

For privacy and ethical reasons, all data are available upon request to the corresponding author.

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
