# Peer review of "Overcoming Pain and Kinesiophobia: Unlocking the Path to Better Knee Osteoarthritis Rehabilitation"

_biomedicines, 2025, doi:10.3390/biomedicines13061361_

Round 1
Reviewer 1 Report
Comments and Suggestions for Authors
Title and Abstract
The current title, while descriptive, may be overly long and repetitive, which can hinder an immediate grasp of the manuscript’s focus. A more concise formulation would highlight the study’s essential themes—pain, kinesiophobia, and knee osteoarthritis—without unnecessary detail. In the abstract, the discussion of cross-lagged panel model coefficients is informative but could be scaled back. Providing a concise overview of the principal finding—that reductions in pain significantly predict decreases in kinesiophobia and increases in quality of life—would be more engaging for clinicians and researchers seeking the main takeaway.
Introduction
Here, the manuscript offers a broad overview of the biopsychosocial (BPS) framework, referencing general health conditions and mortality. Although this background sets the stage for understanding the complexity of knee osteoarthritis, it detracts from the article’s primary focus on KOA-specific factors, such as pain-related anxiety and function. Streamlining or replacing these general references with KOA-focused data—particularly around how chronic pain contributes to fear of movement—would align the introduction more tightly with the study’s aim. Emphasizing the knowledge gap (for instance, that existing literature addresses kinesiophobia and OA, but not in a cross-lagged manner) would also make the study’s novelty clearer.
Methods
The methods section does a solid job of enumerating inclusion and exclusion criteria; however, it raises questions about how treatment protocols were individualized for participants of different ages or disease stages. Offering more in-depth reasoning behind the specific number of treatment sessions (for example, six sessions in total) would provide valuable insight into how you balanced therapeutic efficacy with practical constraints. The article describes Quantum Molecular Resonance (QMR) therapy but leaves the underlying rationale somewhat underexplained: introducing a brief theoretical premise or summarizing previous findings on QMR’s mechanism of action would clarify why this modality is suitable for KOA. Additionally, a short note on electrode placement and dosage parameters—beyond just frequency ranges—would enhance reproducibility for other practitioners who might wish to implement or compare this intervention.
Results
The results section employs cross-lagged panel modeling to tease out directional effects between pain, kinesiophobia, and quality of life, which is commendably thorough from a statistical standpoint. However, the level of detail can overwhelm readers unfamiliar with structural equation modeling. While it is helpful to list models M1 through M4, the main text should emphasize the critical numeric changes (e.g., decreases in VAS, improvements in KOOS) and highlight how these relate directly to the overarching hypothesis (pain reduction driving changes in kinesiophobia). Model-fit indices and path coefficients could be partially shifted to a supplementary table, allowing the main results narrative to remain clear and clinically oriented.
Discussion
In interpreting the findings, the manuscript correctly identifies pain as a major determinant of patient-reported outcomes. Yet it might benefit from clearer statements linking these findings to clinical practice, such as recommending that pain-management interventions be prioritized early in rehabilitation to prevent or reduce kinesiophobia. Additionally, the discussion could further address potential confounders—like the wide age range of participants—that may influence how quickly or effectively fear of movement subsides. A frank acknowledgment of the study’s limitations, such as the short follow-up window and the absence of a comparator or control group, would also enhance the study’s credibility and invite future researchers to build upon these initial results under more controlled conditions.
Conclusion
The concluding remarks appropriately note that alleviating pain can improve both patient autonomy and psychological well-being in knee osteoarthritis; however, adding a concise action-oriented statement would underscore the immediate utility of the findings. For example, specifying that future work should compare QMR with standard physiotherapy in a randomized design over a longer follow-up period would help readers see precisely how the research might evolve. Such a focused conclusion bridges the gap between the current study’s insights and broader clinical or investigative applications, ultimately reinforcing the significance of pain-focused therapies for reducing kinesiophobia in KOA populations.
Reviewer 2 Report
Comments and Suggestions for Authors
Thank you for the opportunity to review your work. Below you will find the main concerns raised:
Introduction
-
Structure and Organization:
- The introduction suffers from uneven paragraph lengths and lacks smooth transitions between ideas. Some paragraphs are excessively long (up to 33 lines), whereas others are very brief (around 3–4 lines). The text would benefit from a thorough reorganization, ensuring each paragraph develops a single coherent idea and transitions logically to the next.
-
Clarity of Rationale:
- The rationale for the study is not sufficiently established. The introduction does not clearly state what is already known about the relationship between pain, kinesiophobia, and quality of life in knee osteoarthritis, nor does it outline the specific knowledge gap that this study intends to fill. The text should clearly delineate what is established in the literature and what remains uncertain.
-
Justification of the Intervention:
- The manuscript does not adequately explain or justify the choice of Quantum Molecular Resonance (QMR) as the intervention. There is no discussion of the mechanism of QMR, its previous applications in musculoskeletal conditions, or why it may be appropriate for knee osteoarthritis. This omission weakens the overall rationale for the study.
-
Consistency in Referencing:
- The citation numbered “19” in the text does not correspond to the correct reference in the reference list. This inconsistency in referencing must be corrected to ensure accuracy and traceability of the cited literature.
Methods
-
Population and Diagnosis:
- The criteria for defining the study population are insufficiently detailed. For knee osteoarthritis, a robust diagnosis should be based on both clinical and radiological findings. The ones chosen are not sufficient to a proper knee osteoarthritis diagnosis (as suggested by Altman et al. 1986 - see: Altman R, Asch E, Bloch D, Bole G, Borenstein D, Brandt K, Christy W, Cooke TD, Greenwald R, Hochberg M, et al. Development of criteria for the classification and reporting of osteoarthritis. Classification of osteoarthritis of the knee. Diagnostic and Therapeutic Criteria Committee of the American Rheumatism Association. Arthritis Rheum. 1986 Aug;29(8):1039-49. doi: 10.1002/art.1780290816. PMID: 3741515).
- Allowing inclusion of patients across three radiographic stages (I, II, and III) without stratification risks introducing significant heterogeneity. It would be preferable to use a more homogeneous sample, for example, focusing predominantly on stage II patients, given that stage I might be clinically insignificant and stage III may present with highly exaggerated signs and symptoms.
-
Intervention Clarification:
- The physical therapy intervention is described as QMR. However, since no other physical therapy modalities (e.g., manual therapy or exercise) were applied, the intervention should be more accurately described as “electrotherapy using QMR.”
- The manuscript must provide a rationale for choosing QMR. A background discussion in the introduction or methods section on the theoretical benefits of QMR—and why it might be effective for knee osteoarthritis—is lacking. A review of the literature to support this choice is essential, particularly since it appears to be little or no previous work applying this technology to this or other musculoskeletal conditions.
-
Statistical Analysis:
- The rationale for using Cross-Lagged Panel Models (CLPMs) is not clearly justified. If the primary aim is to explore the relationship between pain and kinesiophobia, simpler and more traditional statistical tests (e.g., paired t-tests or correlation analysis) could suffice. The authors need to explain why CLPMs are more appropriate for their research questions and what additional insight they provide over traditional methods.
Results
-
Population Description:
- The results section does not include a summary or table describing the study population. A flow diagram showing patient recruitment, inclusion, exclusion, and completion rates should be added to improve clarity and allow readers to assess the representativeness of the sample.
-
Data Presentation:
- The text preceding the tables and figures requires revision. Rather than using evaluative language such as “It is plausible to conclude,” the results section should focus on objectively reporting the findings. Analysis and interpretation should be reserved for the discussion section. The narrative should briefly summarize key data points that are not immediately evident from the tables or figures.
Discussion
-
Depth of Discussion:
- The discussion currently reads more as a summary of results rather than a critical analysis. A robust discussion should compare the findings with those from similar studies, explore potential psychological, physiological, and anatomical mechanisms behind the observed results, and contextualize the findings within the broader literature. The only studies cited were those already explored in the introduction section. No new information was gathered or analysed.
- There is a lack of exploration of why the intervention (QMR) produced the observed changes. The discussion should attempt to explain the underlying mechanisms and discuss how the results align—or conflict—with existing theories and evidence.
-
Instrument Validity and Implications:
- The discussion should address concerns regarding the validity of the instrument(s) used for measuring outcomes, particularly if there is any controversy or lack of validation in the study population.
- Further commentary on the clinical and practical implications of the findings (and any limitations therein) is needed.
Other Aspects
- Ethics Committee:
- The study was reviewed by a psychology ethics committee, which may not be appropriate given that direct electrical interventions were applied to a patient population. It is recommended that the study be reviewed by an ethics committee with expertise in health-related research, particularly in interventions involving physical modalities.
- Conflict of Interest:
- One author is noted as being an employee of the company manufacturing the instrument used in the study. The manuscript should clearly specify its role, because in the Author Contributions it is only abbreviated AP which could be any of the authors Andrea Pantalone, Alessandro Pozzato, or Antonia Patruno.
The entire manuscript requires a thorough language revision. Certain phrases and constructions (e.g., “In other words...” or “We hope that our work...”) are indicative of literal translation and are not standard in high-quality scientific writing. The language should be formal, concise, and clear.
Reviewer 3 Report
Comments and Suggestions for Authors
Thank you for the opportunity to review your manuscript, “The bridge from pain perception to kinesiophobia in knee osteoarthritis in outpatients’ rehabilitation setting: a biopsychological approach”. Overall, the manuscript is well-written and technically correct but needs some little adjustments before acceptance.
Introduction
- You spend a lot of content to introduction what the biopsychosocial approach is. In my opinion, you should simplify the introduction of biopsychosocial approach. You research topic focused on the bridge from pain perception to kinesiophobia in knee OA. You should stated the significance of your research, the state-of-the-art research status, problems in current research, and then you research aim.
- Please state the aim of your research more clearly.
- The hypothesis of your research is missing.
Materials and Methods
- In Line 111, 40 patients were recruited in this study. Did you preformed sample size calculationusing G*Power or other software before the subject recruitment? Whether 40 samples are appropriate for your study?
- In section 2.2,for better understanding your treatment protocol, please add a photo of the real rehabilitative treatments.
- In section 2.3, please add the interpretation of each kinds of scores for better understanding your results in Table 1. For example,what’s the total score of KOOS? What do the total and zero scores mean?
- In section 2.4, the CLPM is a common method for analyzing longitudinal data. Please delete the large interpreted texts about CLPM. It’s too redundant.
- In Line 179, if β3and β4 are significant, X and Y are stable. You mean a significant difference, right? Please modify the text.
- How did you define the significant difference? Which method did you use?
- In Line 181, what is “SEM in R”?
- In Line 183, what are the M1, M2, M3 and M4? Please move the explanation in Line 208 (Result Section) to here.
- In Line 184, what are the AIC, BIC, etc.? please give a simple explanation.
Results
- In the first paragraph of section Results, please use the statistical methods to compare the values in T0, T1 and T3, rather than just comparing the mean values. Please add the specific pvalue for each comparison.
- In Table 1, please add the specific pvalue for each comparison.
- In Line 222, you said that the differences between M2 and M1 and between M4 and M1 are significant. But in Table 2, the p value of M2 vs. M1 was 0.276, and the p value of M4 vs. M1 was 0.075.
- In Line 223, you said that the difference between M3 and M1 is not significant. But in Line 225, you said that the difference between M3 and M1 is significant.
- In Line 224, “It is possible to say that there is a causal predominance of VAS on KOOS both in the first and second wave. ”, “KOOS”should be “TSK”. Please modify it.
- In Line 228, “Figure 3 shows the path diagram of the CLPM that relates VAS measures to TSK ”, the “TSK”should be “KOOS”. Please modify it.
Discussion and Conclusion
- I suggest you separate the conclusion from the
- The discussion section is brief and mainly focuses on summarising the results rather than discussing them.The discussion section would particularly benefit from further elaboration and deeper insights to enhance the quality of the paper.
- In Line 290, you said that the main limitation of your study was the small sample size. The sample size calculation should be performedat the beginning of the study. Maybe 40 samples is appropriate according to the sample size calculation. The problem about the sample size is not your study limitation.
- In Line 293, you said that another limitation of your research is the diverse manifestations of knee OA. The problem should be considered at the beginning of the patient recruitment.
- I suggest you rewrite the study limitations.
- In Line 297, you should stated the specific contribution of your research to clinical work. How do your research findings guide clinical work?
Reviewer 4 Report
Comments and Suggestions for Authors
The manuscript refers to a “single-subject design” model, but the study actually involves 40 participants. This terminology is potentially misleading. Rephrase to “within-subject design” or “longitudinal design” to avoid confusion with N=1 designs.
The rehabilitative protocol using QMR is described, but without sufficient scientific grounding or justification. Include references that validate the effectiveness or physiological mechanisms of QMR in KOA. If this is a novel application, discuss it as a limitation or provide a rationale.
Round 2
Reviewer 1 Report
Comments and Suggestions for Authors
Dear Authors,
Thank you for your detailed and careful revision of the manuscript in response to my previous comments and suggestions. Upon reviewing your revisions and the highlighted amendments, I find that the manuscript has significantly improved in clarity, methodological rigor, and clinical relevance. Specifically, the streamlined title and abstract effectively emphasize the study's core findings, and the adjustments to the introduction now clearly frame the clinical context and research gap related to kinesiophobia in knee osteoarthritis.
The additions and clarifications provided in the Methods section, particularly the further elaboration on the rationale behind Quantum Molecular Resonance (QMR) therapy and the detailed treatment protocol description, notably enhance the reproducibility and transparency of your approach. Furthermore, the introduction of Supplementary Table S1 effectively addresses my previous concerns about the complexity and accessibility of statistical information, thereby making the results section clearer and more clinically meaningful.
Additionally, your adjustments in the Discussion and Conclusion sections successfully link the results to practical clinical implications, acknowledging key study limitations and suggesting avenues for future research, thus strengthening the overall scientific credibility and utility of your work.
Given these substantial improvements, I consider the manuscript suitable and worthy for publication. Congratulations on your rigorous and thoughtful revision.
Best regards,
Author Response
Dear Prof/reviewer,
Thank you for your comments that allowed us to improve our manuscript. Thank you for your approval. Best regards
T. Paolucci
Reviewer 2 Report
Comments and Suggestions for Authors
Dear authors,
Thank you for sending the reviewed version of your manuscript. Although some improvements were made, other issues were maintained.
Comment 1: The introduction suffers from uneven paragraph lengths and lacks smooth transitions between ideas. Some paragraphs are excessively long (up to 33 lines), whereas others are very brief (around 3–4 lines). The text would benefit from a thorough reorganization, ensuring each paragraph develops a single coherent idea and transitions logically to the next.
Response 1: None
Comment 1.1: This issue persists. Please review and address it.
Comment 2: The rationale for the study is not sufficiently established. The introduction does not clearly state what is already known about the relationship between pain, kinesiophobia, and quality of life in knee osteoarthritis, nor does it outline the specific knowledge gap that this study intends to fill. The text should clearly delineate what is established in the literature and what remains uncertain.
Response 2: This kinesiophobia in KOA can become a major barrier in the rehabilitation process because patients tend to avoid activities that might help reduce their symptoms or improve joint function. It has been observed that this fear of movement can create a vicious cycle—leading to inactivity, muscle weakness, and further deterioration of joint function—ultimately worsening pain and disability. Effective rehabilitation strategies focus on reducing this fear through educational approaches and therapeutic interventions, such as cognitive-behavioral therapy and gradual, personalized exercise programs and integrated programs of physical therapy. These methods help patients understand that controlled, progressive movement can be safe and beneficial. Research shows that combining pain management techniques with psychoeducational interventions is crucial in decreasing fear of movement, promoting faster recovery, and improving overall quality of life.”
Comment 2.1: Thank you for your response. While this helps clarify the rationale, the text still lacks proper citations. Even though the authors are specialists in this area, the manuscript would benefit from adding appropriate references.
Comment 3: The manuscript does not adequately explain or justify the choice of Quantum Molecular Resonance (QMR) as the intervention. There is no discussion of the mechanism of QMR, its previous applications in musculoskeletal conditions, or why it may be appropriate for knee osteoarthritis. This omission weakens the overall rationale for the study.
Response 3: In rehabilitation, in musculoskeletal pain and in KOA, instrumental physical therapy is used for pain reduction and as an anti-inflammatory. QRM represents an innovative instrumental physical therapy in this sense. The publications on the subject are reported:
- Paolucci T, Tommasi M, Pozzato G, Pozzato A, Pezzi L, Zuccarini M, Di Lanzo A, Palumbo R, Porto D, Messeri R, Pesce M, Pantalone A, Buda R, Patruno A. Management and Rehabilitative Treatment in Osteoarthritis with a Novel Physical Therapy Approach: A Randomized Control Study. Diagnostics (Basel). 2024 Jun 6;14(11):1200. doi: 10.3390/diagnostics14111200. PMID: 38893726; PMCID: PMC11171699.
- Paolucci T, Pino V, Elsallabi O, Gallorini M, Pozzato G, Pozzato A, Lanuti P, Reis VM, Pesce M, Pantalone A, Buda R, Patruno A. Quantum Molecular Resonance Inhibits NLRP3 Inflammasome/Nitrosative Stress and Promotes M1 to M2 Macrophage Polarization: Potential Therapeutic Effect in Osteoarthritis Model In Vitro. Antioxidants (Basel). 2023 Jun 28;12(7):1358. doi: 10.3390/antiox12071358. PMID: 37507898; PMCID: PMC10376596.
Comment 3.1: Thank you for the response. You showed two studies that demonstrate the use of QMR. Two studies produced by the authors. One study with a vitro design and other with KOA patients. Although the in vitro study highlights potential mechanisms, it has a limited clinical transferability. Futhermore, the second study, although it is an experimental study design in KOA patients, the metodology design have problems, namely the uneven number of participants in the groups (11 vs 20 vs 20), which may introduce statistical bias. Moreover, in the literature, besides those studies, I was unable to find additional studies using this intervention on these patients. Therefore, the concern raised in my previous comment remains unaddressed and the authors did not provide a proper response to the comment.
Comment 4: The citation numbered “19” in the text does not correspond to the correct reference in the reference list. This inconsistency in referencing must be corrected to ensure accuracy and traceability of the cited literature.
Response 4: Done
Comment 4.1: Thank you for updating the references.
Comment 5: The criteria for defining the study population are insufficiently detailed. For knee osteoarthritis, a robust diagnosis should be based on both clinical and radiological findings. The ones chosen are not sufficient to a proper knee osteoarthritis diagnosis (as suggested by Altman et al. 1986 - see: Altman R, Asch E, Bloch D, Bole G, Borenstein D, Brandt K, Christy W, Cooke TD, Greenwald R, Hochberg M, et al. Development of criteria for the classification and reporting of osteoarthritis. Classification of osteoarthritis of the knee. Diagnostic and Therapeutic Criteria Committee of the American Rheumatism Association. Arthritis Rheum. 1986 Aug;29(8):1039-49. doi: 10.1002/art.1780290816. PMID: 3741515). Allowing inclusion of patients across three radiographic stages (I, II, and III) without stratification risks introducing significant heterogeneity. It would be preferable to use a more homogeneous sample, for example, focusing predominantly on-stage II patients, given that stage I might be clinically insignificant and stage III may present with highly exaggerated signs and symptoms.
Response 5: We discuss this point in the discussion “limits of the study “. Furthermore, we availed ourselves of an expert orthopedic surgeon about KOA (with > 20 years of clinical and surgical experience), for the diagnosis, specifying that for the rehabilitation management, the indication for instrumental physical therapy with electromagnetic fields or electrotherapy does not change in the different types of KOA.
Comment 5.1: Thank you for your response. Acknowledging this as a limitation does not explain why more widely accepted KOA diagnostic criteria were not used. Additionally, you still have not addressed the problem of including multiple KOA stages within the groups (increasing the sample size alone does not resolve this heterogeneity).
Comment 6: The physical therapy intervention is described as QMR. However, since no other physical therapy modalities (e.g., manual therapy or exercise) were applied, the intervention should be more accurately described as “electrotherapy using QMR.”
Response 6: None
Comment 6.1: Please review your text accordingly
Comment 7: The manuscript must provide a rationale for choosing QMR. A background discussion in the introduction or methods section on the theoretical benefits of QMR—and why it might be effective for knee osteoarthritis—is lacking. A review of the literature to support this choice is essential, particularly since it appears to be little or no previous work applying this technology to this or other musculoskeletal conditions.
Response 7: In rehabilitation, in musculoskeletal pain and in KOA, instrumental physical therapy is used for pain reduction and as an anti-inflammatory. QRM represents an innovative instrumental physical therapy in this sense. The publications on the subject are reported:
- Paolucci T, Tommasi M, Pozzato G, Pozzato A, Pezzi L, Zuccarini M, Di Lanzo A, Palumbo R, Porto D, Messeri R, Pesce M, Pantalone A, Buda R, Patruno A. Management and Rehabilitative Treatment in Osteoarthritis with a Novel Physical Therapy Approach: A Randomized Control Study. Diagnostics (Basel). 2024 Jun 6;14(11):1200. doi: 10.3390/diagnostics14111200. PMID: 38893726; PMCID: PMC11171699.
- Paolucci T, Pino V, Elsallabi O, Gallorini M, Pozzato G, Pozzato A, Lanuti P, Reis VM, Pesce M, Pantalone A, Buda R, Patruno A. Quantum Molecular Resonance Inhibits NLRP3 Inflammasome/Nitrosative Stress and Promotes M1 to M2 Macrophage Polarization: Potential Therapeutic Effect in Osteoarthritis Model In Vitro. Antioxidants (Basel). 2023 Jun 28;12(7):1358. doi: 10.3390/antiox12071358. PMID: 37507898; PMCID: PMC10376596.
Comment 7.1: Thank you for reiterating your earlier response. Please see my detailed feedback in comment 3.1.
Comment 8: The rationale for using Cross-Lagged Panel Models (CLPMs) is not clearly justified. If the primary aim is to explore the relationship between pain and kinesiophobia, simpler and more traditional statistical tests (e.g., paired t-tests or correlation analysis) could suffice. The authors need to explain why CLPMs are more appropriate for their research questions and what additional insight they provide over traditional methods.
Response 8: Thank you for your insightful comment. While traditional statistical methods such as paired t-tests and correlation analyses can provide valuable information about the association between pain and kinesiophobia at a single point in time or across two time points, they have several limitations in capturing the dynamic, bidirectional relationship between these variables over time. Cross-Lagged Panel Models (CLPMs) offer significant advantages in three key phases of analysis:
- Phase of Temporal Ordering and Directionality: Traditional methods, such as correlation analysis, assess only the concurrent association between pain and kinesiophobia without accounting for temporal precedence. CLPMs, on the other hand, estimate cross-lagged paths, allowing us to determine whether prior levels of pain predict later levels of kinesiophobia and vice versa. This is essential in establishing potential causal directionality in the relationship.
- Phase of Controlling for Stability and Confounding Variables: Simple paired t-tests or correlations do not account for the stability of pain and kinesiophobia over time. That is, they do not separate the influence of past values of each variable on its future values. CLPMs include autoregressive paths that control for prior levels of pain and kinesiophobia, helping to isolate the unique predictive effect of one variable on the other. This ensures that observed associations are not merely due to the persistence of individual differences.
- Phase of Bidirectional Influence and Reciprocal Relationships: Traditional methods struggle to disentangle whether pain influences kinesiophobia, kinesiophobia influences pain, or both interact in a feedback loop. CLPMs explicitly model bidirectional effects, allowing us to determine whether pain precedes kinesiophobia, kinesiophobia precedes pain, or if both have reciprocal influences over time. This provides a deeper understanding of the dynamic interplay between these constructs.
In sum, the choice of CLPMs is justified as they allow for a more rigorous examination of the pain-kinesiophobia relationship by capturing its temporal dynamics, controlling for prior levels of the variables, and identifying potential reciprocal effects.
We inserted a brief sentence in the text that briefly justify our methodological approach.
Comment 8.1: Thank you for your explanation. Although I still believe traditional statistical methods could address the study’s objectives, I now understand and accept your rationale. The statistical analysis is therefore acceptable.
Comment 9: The results section does not include a summary or table describing the study population. A flow diagram showing patient recruitment, inclusion, exclusion, and completion rates should be added to improve clarity and allow readers to assess the representativeness of the sample.
Response 9: None
Comment 9.1: Please review your manuscript accordingly
Comment 10: The text preceding the tables and figures requires revision. Rather than using evaluative language such as “It is plausible to conclude,” the results section should focus on objectively reporting the findings. Analysis and interpretation should be reserved for the discussion section. The narrative should briefly summarize key data points that are not immediately evident from the tables or figures.
Response 10: We changed the language of the text in the result section.
Comment 10.1: Thank you for revising the text.
Comment 11: The discussion currently reads more as a summary of results rather than a critical analysis. A robust discussion should compare the findings with those from similar studies, explore potential psychological, physiological, and anatomical mechanisms behind the observed results, and contextualize the findings within the broader literature. The only studies cited were those already explored in the introduction section. No new information was gathered or analysed.
Comment 12: There is a lack of exploration of why the intervention (QMR) produced the observed changes. The discussion should attempt to explain the underlying mechanisms and discuss how the results align—or conflict—with existing theories and evidence.
Comment 13: The discussion should address concerns regarding the validity of the instrument(s) used for measuring outcomes, particularly if there is any controversy or lack of validation in the study population.
Comment 14: Further commentary on the clinical and practical implications of the findings (and any limitations therein) is needed.
Response 11, 12, 13, and 14: None
Comment 11.1, 12.1, 13.1, and 14.1: While additional content has been added to the discussion section, it does not fully address the issues previously identified. Please revise the manuscript to comprehensively resolve those concerns.
Comment 15: The study was reviewed by a psychology ethics committee, which may not be appropriate given that direct electrical interventions were applied to a patient population. It is recommended that the study be reviewed by an ethics committee with expertise in health-related research, particularly in interventions involving physical modalities
Comment 16: One author is noted as being an employee of the company manufacturing the instrument used in the study. The manuscript should clearly specify its role, because in the Author Contributions it is only abbreviated AP which could be any of the authors Andrea Pantalone, Alessandro Pozzato, or Antonia Patruno.
Response 15 and 16: None
Comment 15.1 and 16.1: These issues must be addressed to clarify the potential for bias in your study. Please provide a detailed response.
Comments on the Quality of English LanguageThe structure and language were not properly improved. Please see my previous comment.
Author Response
attached document

Reviewer 3 Report
Comments and Suggestions for Authors
The author did not respond to all of my previous comments. Many comments were not included in the author response file. Please reply to all of my previous comments one by one. If the author did not agree my comment, please don’t hesitate to tell me.
Author Response
Dear Reviewer,
I report below our answers.
Which point has not been addressed?
Thank you
Kind regards
T. Paolucci
Point by point
Materials and Methods
In Line 111, 40 patients were recruited in this study. Did you preformed sample size calculation using G*Power or other software before the subject recruitment? Whether 40 samples are appropriate for your study?
Authors: The 40 patients were the subjects available because they had to take physiotherapy. G*Power is useful for traditional statistics (as correlation, regression. T-test or Anova) but not for Cross-Lagged Panel Models. The sample size required for a cross-lagged panel model (CLPM) depends on the model's complexity, the reliability of the measures, and the size of the cross-lagged effects. However, in our study, because the population target were people who underwent physiotherapy, the sample size is prevalently determined by the number of available people with this specific characteristic.
In section 2.3, please add the interpretation of each kinds of scores for better understanding your results in Table 1. For example, what’s the total score of KOOS? What do the total and zero scores mean?
Authors: we further specified characteristics of scale score in lines 144-157.
In section 2.4, the CLPM is a common method for analyzing longitudinal data. Please delete the large interpreted texts about CLPM. It’s too redundant.
Authors: we removed the text about the CPLM.
In Line 179, if β3and β4 are significant, X and Y are stable. You mean a significant difference, right? Please modify the text.
Authors: we fixed it
How did you define the significant difference? Which method did you use?
Authors: as we specified in the following line, we used structural equation model implemented with R as method of analysis
In Line 181, what is “SEM in R”?
Authors: structural equation model. We substituted the acronym with the entire expression in the text.
In Line 183, what are the M1, M2, M3 and M4? Please move the explanation in Line 208 (Result Section) to here.
Authors: we fixed it.
In Line 184, what are the AIC, BIC, etc.? please give a simple explanation.
Authors: we specified the meaning of the acronyms
Results
In the first paragraph of section Results, please use the statistical methods to compare the values in T0, T1 and T3, rather than just comparing the mean values. Please add the specific pvalue for each comparison.
In Table 1, please add the specific p-value for each comparison.
Authors: in the first paragraph and in table 1 we reported only descriptives. The CPLM is used to calculate regressions and correlations between phases T0, T1, and T2. The CPLM results are reported in table 2 and 3.
In Line 222, you said that the differences between M2 and M1 and between M4 and M1 are significant. But in Table 2, the p value of M2 vs. M1 was 0.276, and the p value of M4 vs. M1 was 0.075.
Authors: We thank the reviewer for the mistake due to an improper description and we fixed it
In Line 223, you said that the difference between M3 and M1 is not significant. But in Line 225, you said that the difference between M3 and M1 is significant.
Authors: We thank the reviewer for the mistake and we fixed it
In Line 224, “It is possible to say that there is a causal predominance of VAS on KOOS both in the first and second wave. ”, “KOOS”should be “TSK”. Please modify it.
Authors: we fixed it
In Line 228, “Figure 3 shows the path diagram of the CLPM that relates VAS measures to TSK ”, the “TSK”should be “KOOS”. Please modify it.
Authors: we fixed it
Reviewer
Other Aspects
Ethics Committee:
The study was reviewed by a psychology ethics committee, which may not be appropriate given that direct electrical interventions were applied to a patient population. It is recommended that the study be reviewed by an ethics committee with expertise in health-related research, particularly in interventions involving physical modalities.
Author: The approval board of the Department Ethics Committee is not composed only of psychologists, and in addition, depending on the characteristics of the studies, a clinical opinion is asked to experts, as in this case. For greater precision, the work had already been previously approved by the Department of Oral Medical Sciences and Biotechnology of University G. D’Annunzio of Chieti-Pescara. Then, we decided to modified as “ by a local Ethical Review Boards of University G. D’Annunzio (protocol code 22016, 17 December 2022).
Reviewer
Conflict of Interest:
One author is noted as being an employee of the company manufacturing the instrument used in the study. The manuscript should clearly specify its role, because in the Author Contributions it is only abbreviated AP which could be any of the authors Andrea Pantalone, Alessandro Pozzato, or Antonia Patruno.
Dr. A. Pozzato was included among the authors because he actively contributed to the research, in particular to the design of the experiment and to the optimization of the device for specific use. Dr. A. Pozzato was included among the authors because he actively contributed to the research, in particular to the design of the experiment and to the optimization of the device for specific use.
Round 3
Reviewer 2 Report
Comments and Suggestions for Authors
Dear Authors,
Thank you again for your thoughtful responses.
In this round of reviews, you addressed my concerns more clearly. Although some suggested revisions were not fully implemented, I now understand the rationale behind your methodological choices. Additionally, I recognize that certain issues are difficult to resolve once the study has concluded. You can now carry on to the next steps toward publication.
Author Response
Dear Reviewer,
thanks for your comments that have allowed us to improve the manuscript and again, have pushed us to new reflections both with respect to what has already been published on the subject and also, for future studies.
Thanks again
Best Regards
Teresa Paolucci
Reviewer 3 Report
Comments and Suggestions for Authors
Dear authors:
The following comments did not been responded.
Introduction
- You spend a lot of content to introduction what the biopsychosocial approach is. In my opinion, you should simplify the introduction of biopsychosocial approach. You research topic focused on the bridge from pain perception to kinesiophobia in knee OA. You should stated the significance of your research, the state-of-the-art research status, problems in current research, and then you research aim.
- Please state the aim of your research more clearly.
- The hypothesis of your research is missing.
Materials and Methods
2. In section 2.2, for better understanding your treatment protocol, please add a photo of the real rehabilitative treatments.
Discussion and Conclusion
- I suggest you separate the conclusion from the discussion
- The discussion section is brief and mainly focuses on summarising the results rather than discussing them.The discussion section would particularly benefit from further elaboration and deeper insights to enhance the quality of the paper.
- In Line 290, you said that the main limitation of your study was the small sample size. The sample size calculation should be performedat the beginning of the study. Maybe 40 samples is appropriate according to the sample size calculation. The problem about the sample size is not your study limitation.
- In Line 293, you said that another limitation of your research is the diverse manifestations of knee OA. The problem should be considered at the beginning of the patient recruitment.
- I suggest you rewrite the study limitations.
- In Line 297, you should stated the specific contribution of your research to clinical work. How do your research findings guide clinical work?
Author Response
Introduction
- You spend a lot of content to introduction what the biopsychosocial approach is. In my opinion, you should simplify the introduction of biopsychosocial approach. You research topic focused on the bridge from pain perception to kinesiophobia in knee OA. You should stated the significance of your research, the state-of-the-art research status, problems in current research, and then you research aim.
- Please state the aim of your research more clearly.
- The hypothesis of your research is missing.
Authors: Dear Reviewer, thanks. We modified respect to your suggestions:
We cancelled : " Over the past forty years, the principles of the BPS approach have gained renewed interest among healthcare researchers and practitioners Evidence increasingly shows that the quality of social relationships plays a crucial role in ensuring a high level of welfare and mortality in different diseases [3,4,5].; In addition to social factors, psychological characteristics significantly influence health outcomes. For example, studies have demonstrated that feelings of hopelessness can predict the risk of myocardial infarction and early mortality [4, 5]. ; Recognizing these interconnections underscores the importance of a multidisciplinary approach to understanding human health, one that considers all these aspects to better predict risks and tailor interventions.;
Also, .........Our hypothesis considers that kinesiophobia could act as a significant barrier to effective rehabilitation—leading patients to avoid activities that could alleviate symptoms or improve joint function— than, it is crucial to examine how these factors influence each other over time. To better understand these complex interactions, our study focused on analyzing how perceived pain (a biological factor), kinesiophobia (a psychological factor), and quality of life (a social and life condition factor) change during a specific rehabilitative treatment for KOA outpatients.
Materials and Methods
2. In section 2.2, for better understanding your treatment protocol, please add a photo of the real rehabilitative treatments.
Authors: Dear Reviewer, we referred to the previously published manuscript ( Randomized Control Study ) on validation for QRM in KOA, where the protocol is described in more detail and there are reference photos.
- 24. Paolucci T, Tommasi M, Pozzato G, Pozzato A, Pezzi L, Zuccarini M, Di Lanzo A, Palumbo R, Porto D, Messeri R, Pesce M, Pantalone A, Buda R, Patruno A. Management and Rehabilitative Treatment in Osteoarthritis with a Novel Physical Therapy Approach: A Randomized Control Study. Diagnostics (Basel). 2024 Jun 6;14(11):1200. doi: 10.3390/diagnostics14111200. PMID: 38893726; PMCID: PMC11171699.
Discussion and Conclusion
- I suggest you separate the conclusion from the discussion
AUTHORS: Authors: Dear Reviewer, done
- The discussion section is brief and mainly focuses on summarising the results rather than discussing them.The discussion section would particularly benefit from further elaboration and deeper insights to enhance the quality of the paper.
AUTHORS: Authors: Dear Reviewer, done
- In Line 290, you said that the main limitation of your study was the small sample size. The sample size calculation should be performed at the beginning of the study. Maybe 40 samples is appropriate according to the sample size calculation. The problem about the sample size is not your study limitation.
AUTHORS: Authors: Dear Reviewer, we removed this part
- In Line 293, you said that another limitation of your research is the diverse manifestations of knee OA. The problem should be considered at the beginning of the patient recruitment.
AUTHORS: Authors: Dear Reviewer, we removed this part in some points; We had to leave out other points because they were requested by other reviewers.
"While the number of outpatients with KOA varies based on their willingness to participate in rehabilitation programs, it holds that a larger participant pool improves the generalizability of scientific findings. Another limitation concerns the variable manifestations of KOA; these defferences are likely to influence pain levels and psychological well-being disparately, considering individual factors such as age, comorbidity with other illnesses, socio-economic status, and psychological traits like personality or self-efficacy." ; Also, having a larger sample size available it would be useful to stratify the data according to the classification for Kellegren-Lawrence staging score (I-II-III).
- I suggest you rewrite the study limitations.
AUTHORS: Authors: Dear Reviewer, we rewrote the study limitations.
- In Line 297, you should stated the specific contribution of your research to clinical work. How do your research findings guide clinical work?
AUTHORS: Authors: Dear Reviewer, we modified this part as you suggest in line with the requests of the other three reviewers. Thank you for your suggestions and comments that helped us improve the manuscript.
"
Effectively reducing pain not only improves physical function but also decreases fear of movement, which can lead to better psychological and social outcomes. The persistent relationship between pain and kinesiophobia, even during follow-up, suggests that alleviating chronic pain is crucial for reducing anxiety, stress, and ultimately enhancing overall quality of life.
Furthermore, our results emphasize the need for an integrated, biopsychosocial approach—combining pain management techniques with strategies like awareness training and cognitive-behavioral therapy. Such approaches can help patients overcome fear of movement, encouraging active participation in therapy and improving functional outcomes."
Best Regards
Teresa Paolucci

Round 4
Reviewer 3 Report
Comments and Suggestions for Authors
The article revised as required.